# Moderation of the real-world effectiveness of smoking cessation aids by mental health conditions: A population study

**Sarah E. Jackson**[1,2]*, **Leonie Brose**[2,3], **Vera Buss**[1,2], **Lion Shahab**[1,2], **Deborah Robson**[2,3], **Jamie Brown**[1,2]

**1** Department of Behavioural Science and Health, University College London, London, United Kingdom,
**2** SPECTRUM Consortium, London, United Kingdom, **3** Addictions Department, Institute of Psychiatry, Psychology and Neuroscience, King's College London, London, United Kingdom

* s.e.jackson@ucl.ac.uk

**Data Availability Statement:** Data used in these analyses are available on Open Science Framework (https://osf.io/5xubc/).

## Abstract

### Objective

To examine whether the real-world effectiveness of popular smoking cessation aids differs between users with and without a history of mental health conditions.

### Design

Nationally-representative cross-sectional survey conducted monthly between 2016–17 and 2020–23.

### Setting

England.

### Participants

5,593 adults (2,524 with a history of ≥1 mental health conditions and 3,069 without) who had smoked regularly within the past year and had attempted to quit at least once in the past year.

### Main outcome measures

The outcome was self-reported abstinence from quit date up to the survey. Independent variables were use of the following cessation aids during the most recent quit attempt: prescription nicotine replacement therapy (NRT), NRT over-the-counter, varenicline, bupropion, vaping products, face-to-face behavioural support, telephone support, written self-help materials, websites, hypnotherapy, Allen Carr's Easyway, heated tobacco products, and nicotine pouches. The moderator was history of diagnosed mental health conditions (yes/no). Covariates included sociodemographic characteristics, level of cigarette addiction, and characteristics of the quit attempt.

**Funding:** This work was supported by CRUK (PRCRPG-Nov21\100002 to JB) and UK Prevention Research Partnership (MR/S037519/1 to LB). The funders had no role in study design, data collection and analysis, decision to publish, or preparation of the manuscript.

**Competing interests:** We have read the journal's policy and the authors of this manuscript have the following competing interests: JB has received unrestricted research funding from Pfizer and J&J, who manufacture smoking cessation medications. LS has received honoraria for talks, an unrestricted research grant and travel expenses to attend meetings and workshops from Pfizer, and has acted as paid reviewer for grant awarding bodies and as a paid consultant for health care companies. All authors declare no financial links with tobacco companies, e-cigarette manufacturers, or their representatives. There are no patents, products in development or marketed products associated with this research to declare. This does not alter our adherence to PLOS ONE policies on sharing data and materials.

## Results

Relative to those without, participants with a history of mental health conditions were significantly more likely to report using vaping products (38.8% [95%CI 36.7–40.8] vs. 30.7% [28.9–32.5]), prescription NRT (4.8% [3.9–5.7] vs. 2.7% [2.1–3.3]), and websites (4.0% [3.2–4.8] vs. 2.2% [1.6–2.7]). Groups did not differ significantly in their use of other aids. After adjusting for covariates and use of other cessation aids, those who used vaping products (OR = 1.92, 95%CI 1.61–2.30), varenicline (OR = 1.88, 95%CI 1.19–2.98), or heated tobacco products (OR = 2.33, 95%CI 1.01–5.35) had significantly higher odds of quitting successfully than those who did not report using these aids. There was little evidence that using other cessation aids increased the odds of successful cessation, or that the user's history of mental health conditions moderated the effectiveness of any aid.

## Conclusions

Use of vaping products, varenicline, or heated tobacco products in a quit attempt was associated with significantly greater odds of successful cessation, after adjustment for use of other cessation aids and potential confounders. There was no evidence to suggest the effectiveness of any popular cessation aid differed according to the user's history of mental health conditions.

## Introduction

Tobacco smoking remains a leading cause of preventable illness and premature mortality in England [1]. Relative to the general population, people with mental health conditions are more likely to smoke, smoke more heavily, and show greater signs of dependence [2–5]. They are also at increased risk of tobacco-related morbidities, including cardiovascular disease [6, 7], which causes them to have substantially lower life expectancy [8, 9]. Quitting smoking can reduce these risks [10]. A range of smoking cessation aids have been found to increase successful smoking cessation in randomised controlled trials (RCTs) [11–14] and in real-world settings [15–24]. Understanding whether and, if so, how far their effectiveness differs between people with and without mental health conditions can help health professionals and patients to make informed choices around the use of aids for smoking cessation.

In England, a comprehensive range of smoking cessation medications and behavioural support are available [25]. Pharmacological aids include nicotine replacement therapy (NRT), which is available free of charge on prescription or can be bought over-the-counter (OTC), and varenicline (Champix) and bupropion (Zyban) which are only available on prescription. The supply of Champix, the most effective of these [14, 23, 24], was disrupted in 2021 due to nitrosamine impurities found by its supplier, Pfizer [26]. The supply of Zyban was disrupted in late 2022 due to similar concerns about nitrosamine impurities found by its supplier, GSK [27]. Both medications remain unavailable as of March 2024. Generic versions of varenicline are available in other countries and cytisine, a drug with similar properties to varenicline, is licensed [28] and has begun to be supplied in England since January 2024.

Vaping products (often referred to as e-cigarettes) and nicotine pouches are available from specialist 'vape shops', supermarkets, smaller convenience stores, and online [29]. Heated tobacco products have been available in the UK since 2016 but their efficacy for smoking cessation is uncertain [30]. Smokers also have access to free dedicated stop smoking services, which

offer behavioural support, pharmacotherapy, and in some cases, vaping products [31]. Telephone support is available via a free Smokefree National Helpline, and websites offer information on quitting, other forms of support available, and how to access them. Other behavioural treatments, including hypnotherapy and Allen Carr's Easyway method (a single-session pharmacotherapy-free behavioural programme) [32, 33], are provided by private companies.

Around one in two attempts to stop smoking in England involves the use of at least one of these cessation aids [34]. Vaping products are most commonly used (~30% of quit attempts), followed by NRT available OTC (~10%) and medications obtained on prescription (NRT, varenicline, or bupropion; ~5%) [34].

It is possible that the effectiveness of these smoking cessation aids may differ between people with and without mental health conditions [35]. People with mental health conditions may experience stronger reinforcing effects of nicotine, more severe withdrawal symptoms when they try to quit, and greater cessation fatigue (being tired of trying to stop smoking) [36]. As such, it is possible that they may benefit more from cessation aids that mimic the effect of nicotine (e.g., varenicline) or provide an alternative source of nicotine (e.g., NRT, vaping products, nicotine pouches) and less from other cessation aids (e.g., forms of behavioural support). On the other hand, people with mental health conditions may be less likely to adhere to treatments [37], causing effectiveness to be lower.

To our knowledge, just two large experimental studies have investigated whether the effectiveness of smoking cessation treatments is moderated by a person's mental health status. These have focused on varenicline, bupropion, and NRT. One large RCT ('EAGLES') compared varenicline and bupropion with nicotine patch and placebo and showed similar efficacy of these medications for smokers with and without psychiatric disorders [38]. However, a recent secondary analysis of another RCT that compared the effectiveness of bupropion and varenicline reported a slightly different pattern of results [39]. While varenicline was associated with similar quitting outcomes for smokers with depressive symptoms than those without, bupropion appeared to be *less* effective as a smoking cessation aid for those with depressive symptoms [39]. Further research is required on these and other cessation aids. Observational data can shed light on any differences in treatment effectiveness in real-world settings [40].

Using data from the Smoking Toolkit Study, a large, nationally-representative survey of adults in England, this study aimed to comprehensively examine whether the real-world effectiveness of popular smoking cessation aids differs between users with and without mental health conditions. Data on smoking status in relation to history of mental health conditions have been published elsewhere [2], so this paper focused specifically on the use and effectiveness of different cessation aids among those attempting to quit smoking. Specifically, we aimed to address the following research questions:

1. To what extent does a history of one or more diagnosed mental health conditions moderate associations between use (vs. non-use) of various cessation aids in a quit attempt and chances of success?

2. Are any moderating effects similar for those with a single mental health condition and those with multiple mental health conditions?

## Materials and methods

### Pre-registration

The study protocol and analysis plan were pre-registered on Open Science Framework (osf.io/5xubc). We made one amendment. We had planned to calculate Bayes factors for non-

significant interactions based on an expected effect size of OR = 1.5 in the observed direction (i.e., OR = 1.5 for observed ORs >1 and OR = 0.67 for observed ORs <1). Instead, we calculated Bayes factors in both directions, to offer more insight into whether the data suggested a given aid was more or less effective for people with a history of mental health conditions than those without.

## Design

This was an observational study using data from the Smoking Toolkit Study; a nationally-representative monthly cross-sectional survey of adults (≥16 years) in England [41]. The study uses a hybrid of random probability and simple quota sampling to select a new sample of approximately 1,700 adults aged ≥16 years each month. Comparisons with other national surveys and sales data indicate that the survey obtains nationally-representative estimates for key variables including sociodemographic characteristics, smoking prevalence, and cigarette consumption [41, 42]. The Smoking Toolkit Study is coordinated by this study's authors at University College London (PI Jamie Brown).

Data collection for the Smoking Toolkit Study began in November 2006 and the study continues to collect data from a new sample each month. Up to February 2020, the survey was conducted via face-to-face computer-assisted interviews. However, social distancing restrictions introduced during the Covid-19 pandemic meant that no data were collected in March 2020, and data from April 2020 onwards have been collected via telephone interviews. The telephone interviews use a similar sampling and weighting approach as the face-to-face interviews and data collected via the two modalities show good comparability [43–45]. Data were not collected from 16 and 17 year olds between April 2020 and December 2021.

While a core set of questions is included in each monthly survey, other variables are only assessed in certain waves, depending on availability of competitive research funding. Questions on mental health have been collected in two periods: January 2016-December 2017 and October 2020-June 2023. We used data from participants surveyed in these periods. We selected participants aged ≥18 years who:

i.  smoked cigarettes (including hand-rolled) or any other tobacco product (e.g., pipe or cigar) daily or occasionally at the time of the survey or during the past year; and

ii.  reported having made at least one serious quit attempt in the past year.

## Ethics statement

Ethical approval was provided by the UCL Research Ethics Committee (0498/001). Participants provided informed verbal consent to take part in the study, and all methods are carried out in accordance with relevant regulations. The data are not collected by UCL and are anonymised when received by UCL.

## Measures

**Outcome: Successful smoking cessation.**   The outcome variable was self-reported continuous abstinence from the start of the most recent quit attempt up to the time of survey. Respondents were asked 'How long did your most recent quit attempt last before you went back to smoking?' Responses were coded 1 for those who responded that they were still not smoking and 0 otherwise.

**Exposures: Use of cessation aids.**   Independent variables were self-reported use or not (dummy coded) of the following smoking cessation aids in the most recent quit attempt:

prescription NRT (available in England from prescribing health professionals, including advisors at specialist stop smoking services); NRT available OTC; varenicline; bupropion; vaping products; face-to-face behavioural support; telephone support; written self-help materials; websites; hypnotherapy; Allen Carr's Easyway; heated tobacco products; and nicotine pouches.

Respondents were asked to indicate all that apply, and data for each were coded 1 if chosen and 0 if not. Heated tobacco products were included in the list of response options from April 2016 and nicotine pouches from June 2021; given the low prevalence of use of these products [46, 47], we imputed missing values as 0 for participants surveyed before the response options were introduced.

**Moderator: History of mental health conditions.** Diagnosed mental health conditions were assessed with the question: 'Since the age of 16, which of the following, if any, has a doctor or health professional ever told you that you had?' followed by a list of ICD-10 recognised conditions: depression; anxiety; obsessive compulsive disorder; panic disorder or a phobia; post-traumatic stress disorder; psychosis; personality disorder; attention deficit hyperactivity disorder; an eating disorder; alcohol misuse or dependence; drug use or dependence; and problem gambling. Between 2020 and 2023, this list also included: autism or autism spectrum disorder; and bipolar disorder.

For our primary analysis (RQ1), those who reported any of these diagnoses were coded 1, else they were coded 0 (including those who did not respond, responded 'don't know', or 'prefer not to say').

For our secondary analysis (RQ2), we subdivided the group reporting mental health diagnoses to create a three-level variable: no history of mental health conditions (coded 0), single mental health condition (1 diagnosis; coded 1), and multiple mental health conditions ($\geq 2$ diagnoses; coded 2), given previous evidence showing stronger associations with smoking outcomes among those with multiple conditions [2].

**Covariates.** Covariates included a range of sociodemographic characteristics, level of cigarette addiction, and variables relating to the most recent quit attempt.

Sociodemographic covariates included age, gender, and occupational social grade (ABC1, which includes managerial, professional and intermediate occupations, vs. C2DE, which includes lower supervisory and technical occupations, semi-routine and routine occupations, never worked and long-term unemployed).

Level of cigarette addiction was assessed by asking participants to self-report ratings of the strength of urges to smoke over the last 24 hours (not at all (coded 0), slight (1), moderate (2), strong (3), very strong (4), extremely strong (5)). This question was also coded '0' for smokers who respond 'not at all' to the (separate) question 'How much of the time have you spent with the urge to smoke?' [48]. This validated measure has similar predictive value as the Fagerström Test of Cigarette Dependence and the Heaviness of Smoking Index in for cessation [49].

The characteristics of the most recent quit attempt included time since the quit attempt started (<1 month, 1–6 months, >6 months), the number of prior quit attempts in the past year (1, 2, 3 or $\geq 4$), whether the quit attempt was planned, and whether the respondent cut down first or stopped abruptly.

The month and year of survey were also included to account for seasonal variation in quit attempts (e.g., in January or 'Stoptober' [50, 51]) and changes in the availability and regulation of different smoking cessation aids over the study period. We also adjusted for the mode of data collection with a variable coded 0 up to February 2020 (when data were collected face to face) and 1 from April 2020 onwards (when data were collected via telephone).

## Statistical analysis

The Smoking Toolkit Study uses survey weights to adjust data so that the sample matches the demographic profile of England on age, social grade, region, housing tenure, ethnicity and working status within sex [41]. The following analyses used weighted data. Missing values were excluded on a per-analysis basis.

We calculated the proportion and 95% confidence interval (CI) of smokers with and without mental health conditions reporting using each cessation aid in the most recent quit attempt, and the quit success rate among users of each aid. We also provided descriptive data on the proportion reporting using each cessation aid and the overall quit success rate separately for each individual mental health condition.

We used logistic regression to analyse associations between self-reported abstinence (abstinent yes vs. no) and use of different smoking cessation aids (use of a specific aid vs. no use of that specific aid), adjusting for mental health status, covariates, and use of other cessation aids (baseline model). We repeated the baseline model with the addition of the two-way interaction between mental health diagnoses (0 vs ≥1 mental health conditions) and each cessation aid in turn.

To explore any differences between those with single and multiple mental health diagnoses, we reran the interactions using a 3-level mental health variable (0, 1, ≥2 mental health conditions).

We calculated Bayes factors using an online calculator (bayesfactor.info) to aid in the interpretation of non-significant interactions with mental health diagnoses. These enabled us to examine whether these associations could best be characterised as evidence of no effect, evidence of an effect, or whether data were insensitive to detect an effect [52, 53]. Alternative hypotheses were represented by half-normal distributions and the expected effect size set to OR = 1.5 in the observed direction (OR = 1.5 where the observed OR was >1 and OR = 0.67 when the observed OR was <1) [23].

## Results

Of 95,952 participants surveyed in eligible waves, 17,394 (18.1%) reported smoking in the past year, of whom 5,741 (33.0%) attempted to stop smoking in the past year. We excluded 148 participants with missing data on mental health conditions (there were no missing data on use of cessation aids), leaving a final sample for analysis of 5,593 participants.

Just under half (45.1% weighted) of participants reported having ever been diagnosed with a mental health condition; 16.8% reported a single mental health condition and 28.3% multiple conditions. **Table 1** summarises the characteristics of participants with and without a history of mental health conditions. **S1 Table** presents corresponding data for those reporting single and multiple mental health conditions. Relative to those without, participants reporting a history of mental health conditions were more likely to be younger, identify as women or nonbinary, and come from less advantaged social grades. They also reported a higher level of cigarette addiction, on average, but there were no notable differences in the characteristics of their most recent quit attempt.

**Table 2** summarises use of cessation aids by participants' history of mental health conditions. Participants with a history of mental health conditions were significantly more likely to report using one or more cessation aids (58.9%) than those without (52.9%) and to report using multiple aids (12.0% vs. 7.3%). Among those with and without a history of mental health conditions, vaping products were the most commonly used aid (38.8% and 30.7%, respectively), followed by NRT available over-the-counter (16.7% and 17.8%). All other aids were used by <5% of participants. Relative to those without, participants with a history of mental health conditions were more likely to report using vaping products, prescription NRT, and websites. Use of other aids did not differ significantly between groups. There was no significant

**Table 1. Weighted sample characteristics.**

|  | All participants | No history of a MHC | One or more MHCs |
|---|---|---|---|
| *Unweighted N* | *5593* | *3069* | *2524* |
| **Sociodemographic characteristics** |  |  |  |
| Age (years), % |  |  |  |
| 16–24 | 21.4 (20.2–22.6) | 19.4 (17.9–21.0) | 23.8 (22.0–25.6) |
| 25–34 | 27.7 (26.4–29.0) | 26.5 (24.7–28.3) | 29.2 (27.3–31.2) |
| 35–44 | 18.4 (17.3–19.6) | 18.4 (16.9–20.0) | 18.4 (16.8–20.2) |
| 45–54 | 15.0 (14.0–16.0) | 15.7 (14.4–17.1) | 14.1 (12.8–15.6) |
| 55–64 | 10.4 (9.6–11.2) | 11.0 (10.0–12.2) | 9.5 (8.5–10.8) |
| ≥65 | 7.2 (6.5–7.9) | 9.0 (8.1–10.0) | 4.9 (4.1–5.8) |
| Gender, % |  |  |  |
| Man | 52.2 (50.8–53.6) | 59.8 (57.9–61.7) | 42.9 (40.8–45.0) |
| Woman | 47.0 (45.6–48.4) | 40.0 (38.1–41.8) | 55.6 (53.5–57.7) |
| In another way[1] | 0.8 (0.6–1.1) | 0.2 (0.1–0.5) | 1.6 (1.2–2.1) |
| Social grade, % |  |  |  |
| ABC1 (more advantaged) | 43.2 (41.8–44.6) | 46.0 (44.1–47.8) | 39.8 (37.9–41.8) |
| C2DE (less advantaged) | 56.8 (55.4–58.2) | 54.0 (52.2–55.9) | 60.2 (58.2–62.1) |
| **Level of cigarette addiction** |  |  |  |
| Strength of urges to smoke, mean (SD) | 1.7 (1.3) | 1.5 (1.2) | 1.8 (1.3) |
| **Features of the most recent quit attempt** |  |  |  |
| Time since quit attempt started, % |  |  |  |
| < 1 month | 15.6 (14.6–16.6) | 16.0 (14.6–17.4) | 15.1 (13.6–16.6) |
| Between 1 and 6 months | 46.5 (45.1–47.9) | 45.5 (43.6–47.4) | 47.7 (45.6–49.9) |
| > 6 months | 37.9 (36.6–39.3) | 38.5 (36.7–40.4) | 37.2 (35.2–39.3) |
| Number of past-year quit attempts, % |  |  |  |
| 1 | 64.7 (63.3–66.0) | 65.0 (63.1–66.8) | 64.4 (62.3–66.4) |
| 2 | 20.2 (19.1–21.4) | 20.3 (18.8–21.9) | 20.1 (18.4–21.8) |
| 3 | 7.7 (7.0–8.5) | 7.2 (6.3–8.3) | 8.2 (7.2–9.4) |
| ≥4 | 7.4 (6.7–8.2) | 7.5 (6.5–8.6) | 7.3 (6.3–8.5) |
| Quit attempt was unplanned, % | 56.6 (55.1–58.0) | 56.4 (54.5–58.4) | 56.7 (54.6–58.8) |
| Quit attempt was abrupt, % | 53.4 (52.0–54.8) | 53.1 (51.1–55.0) | 53.8 (51.7–55.9) |

MHC, mental health condition.

Data are weighted to match the adult population in England.

Note: There were some missing data for the following variables: age *n* = 3, sex *n* = 9, strength of urges to smoke *n* = 77, time since quit attempt started *n* = 69, quit attempt was unplanned *n* = 179, quit attempt was abrupt *n* = 80. Valid percentages are shown.

[1] This group was excluded from the regression analyses (which adjust for gender) in Table 3 due to low numbers.

Corresponding data with history of MHCs coded as 0, 1, or ≥2 MHCs are provided in S1 Table.

difference in the prevalence of use of any cessation aid between those with single versus multiple mental health conditions (S2 Table).

Table 2 also shows unadjusted quit success rates for those using each cessation aid. Although absolute differences appeared large for some aids (e.g., written self-help materials and nicotine pouches), wide confidence intervals meant there was no statistically significant difference in quit rates between users of aids with and without a history of mental health conditions, before adjustment for potential confounding variables and use of other aids.

Table 3 shows the results of the logistic regression analyses. The baseline model, which included use (vs. non-use) of each cessation aid, history of mental health conditions, and

**Table 2. Use of cessation aids in the most recent quit attempt by history of mental health conditions.**

| | % (95% CI) | | |
|---|---|---|---|
| | **All participants** | **No history of a MHC** | **One or more MHCs** |
| **Use in the most recent quit attempt of...[1]** | | | |
| Vaping products | 34.3 (33.0–35.7) | 30.7 (28.9–32.5) | 38.8 (36.7–40.8) |
| NRT available over-the-counter | 17.3 (16.2–18.4) | 17.8 (16.4–19.3) | 16.7 (15.1–18.2) |
| Prescription NRT | 3.7 (3.1–4.2) | 2.7 (2.1–3.3) | 4.8 (3.9–5.7) |
| Varenicline | 3.5 (3.0–4.0) | 3.5 (2.8–4.3) | 3.4 (2.7–4.2) |
| Websites | 3.0 (2.5–3.5) | 2.2 (1.6–2.7) | 4.0 (3.2–4.8) |
| Face-to-face behavioural support | 2.2 (1.8–2.7) | 1.8 (1.3–2.3) | 2.8 (2.1–3.4) |
| Allen Carr's Easyway | 1.3 (0.9–1.6) | 1.2 (0.7–1.6) | 1.4 (0.9–1.9) |
| Written self-help materials | 0.9 (0.6–1.2) | 1.1 (0.6–1.5) | 0.7 (0.4–1.1) |
| Nicotine pouches | 0.9 (0.6–1.2) | 0.8 (0.4–1.1) | 1.0 (0.6–1.5) |
| Telephone support | 0.8 (0.5–1.0) | 0.7 (0.4–1.0) | 0.9 (0.5–1.3) |
| Heated tobacco products | 0.8 (0.5–1.0) | 0.7 (0.3–1.0) | 0.9 (0.5–1.2) |
| Hypnotherapy | 0.7 (0.5–0.9) | 0.6 (0.3–0.9) | 0.8 (0.4–1.1) |
| Bupropion | 0.5 (0.3–0.7) | 0.4 (0.2–0.6) | 0.6 (0.2–0.9) |
| **Number of these aids used** | | | |
| 0 (unaided quitting) | 44.4 (43.0–45.8) | 47.1 (45.2–49.0) | 41.1 (39.1–43.2) |
| 1 | 45.8 (44.4–47.2) | 44.9 (43.0–46.8) | 46.9 (44.8–49.0) |
| 2 | 7.3 (6.6–8.1) | 6.2 (5.3–7.2) | 8.7 (7.6–10.0) |
| 3 or more | 2.5 (2.1–3.0) | 1.9 (1.4–2.5) | 3.3 (2.6–4.1) |
| **Quit success among those who used...** | | | |
| Vaping products | 26.1 (23.9–28.2) | 24.8 (21.7–27.8) | 27.4 (24.3–30.4) |
| NRT available over-the-counter | 19.8 (17.1–22.6) | 19.2 (15.6–22.8) | 20.7 (16.5–24.9) |
| Prescription NRT | 21.0 (14.6–27.4) | 21.6 (10.5–32.8) | 20.6 (12.9–28.2) |
| Varenicline | 22.0 (15.3–28.6) | 21.5 (12.0–30.9) | 22.6 (13.3–31.9) |
| Websites | 26.0 (18.5–33.6) | 20.4 (9.5–31.3) | 29.7 (19.4–40.0) |
| Face-to-face behavioural support | 22.7 (14.5–31.0) | 20.0 (8.0–32.0) | 25.0 (13.5–36.4) |
| Allen Carr's Easyway | 15.2 (5.9–24.6) | 13.3 (1.7–25.0) | 17.2 (2.2–32.1) |
| Written self-help materials | 14.5 (3.9–25.1) | 10.9 (0–23.5) | 20.8 (0.8–40.8) |
| Nicotine pouches | 24.7 (11.4–38.0) | 18.2 (1.7–34.7) | 30.5 (9.9–51.0) |
| Telephone support | 31.5 (16.7–46.3) | 36.4 (12.6–60.1) | 27.3 (8.0–46.6) |
| Heated tobacco products | 27.4 (13.4–41.3) | 29.3 (9.1–49.4) | 25.7 (5.0–46.4) |
| Hypnotherapy | 28.0 (13.3–42.8) | 22.6 (2.7–42.5) | 33.0 (10.6–55.4) |
| Bupropion | 27.0 (7.9–46.2) | 30.1 (2.5–57.8) | 24.4 (0–53.3) |
| None of these (unaided quitting) | 21.4 (19.7–23.2) | 23.2 (20.8–25.6) | 18.9 (16.4–21.5) |

MHC, mental health condition. NRT, nicotine replacement therapy.

Data are weighted to match the adult population in England.

[1] Sorted by prevalence of use among all participants in the sample (highest-lowest). Note that response options were not mutually exclusive and prevalence estimates across the different aids therefore do not sum to 100%.

Corresponding data with history of MHCs coded as 0, 1, or ≥2 MHCs are provided in S2 **Table**. Data on use of cessation aids and overall quit success rates by individual MHC are provided in **S3 Table**.

covariates, indicated participants who used vaping products (OR = 1.92, 95%CI 1.61–2.30), varenicline (OR = 1.88, 95%CI 1.19–2.98), or heated tobacco products (OR = 2.33, 95%CI 1.01–5.35) in their quit attempt had significantly higher odds of quitting successfully than those who did not. Those who used Allen Carr's Easyway method (either via face-to-face session [19.8%], book [66.9%], or both [13.3%]) had significantly lower odds of quitting

**Table 3. Real-world effectiveness of cessation aids for successful smoking cessation and interactions with the user's history of mental health conditions.**

| Use in the most recent quit attempt of. . . | Main effect of aid use[1] | | Interaction between aid use and history of MHCs (≥1 vs. 0 MHCs)[2] | | | |
|---|---|---|---|---|---|---|
| | OR (95% CI) | p | OR (95% CI) | p | BF, less effective (OR = 0.67)[3] | BF, more effective (OR = 1.5)[3] |
| Vaping products | 1.92 (1.61–2.30) | <0.001 | 1.25 (0.88–1.78) | 0.213 | 0.20 | 1.34 |
| NRT available over-the-counter | 1.24 (0.97–1.58) | 0.088 | 0.92 (0.57–1.48) | 0.726 | 0.67 | 0.41 |
| Prescription NRT | 1.22 (0.79–1.90) | 0.372 | 0.50 (0.20–1.21) | 0.124 | 2.13 | 0.38 |
| Varenicline | 1.88 (1.19–2.98) | 0.007 | 1.14 (0.47–2.77) | 0.772 | 0.65 | 0.87 |
| Websites | 1.23 (0.74–2.05) | 0.428 | 1.46 (0.50–4.24) | 0.487 | 0.60 | 1.16 |
| Face-to-face behavioural support | 1.08 (0.59–2.00) | 0.805 | 0.81 (0.24–2.77) | 0.740 | 0.98 | 0.73 |
| Allen Carr's Easyway | 0.41 (0.18–0.94) | 0.034 | 0.69 (0.13–3.66) | 0.665 | 1.06 | 0.78 |
| Written self-help materials | 0.50 (0.19–1.34) | 0.166 | 0.54 (0.09–3.34) | 0.510 | 1.06 | 0.75 |
| Nicotine pouches | 1.08 (0.52–2.24) | 0.838 | 2.36 (0.53–10.5) | 0.261 | 0.61 | 1.43 |
| Telephone support | 1.51 (0.64–3.57) | 0.353 | 0.70 (0.14–3.51) | 0.659 | 1.05 | 0.77 |
| Heated tobacco products | 2.33 (1.01–5.35) | 0.047 | 0.46 (0.09–2.50) | 0.371 | 1.26 | 0.68 |
| Hypnotherapy | 0.82 (0.37–1.81) | 0.623 | 1.89 (0.37–9.71) | 0.448 | 0.71 | 1.20 |
| Bupropion | 1.59 (0.45–5.68) | 0.474 | 0.76 (0.07–8.79) | 0.827 | 1.01 | 0.90 |

BF, Bayes factor. MHC, mental health condition. NRT, nicotine replacement therapy.

Data are weighted to match the adult population in England.

[1] Baseline model, adjusted for use of other cessation aids, history of MHCs (0 vs. ≥1), age, gender, occupational social grade, strength of urges to smoke, time since the most recent quit attempt started, number of past-year quit attempts, whether the quit attempt was planned, whether the quit attempt was abrupt or gradual, and survey month and year.

[2] Baseline model with the addition of the two-way interaction between use of the aid of interest and history of MHCs (0 vs. ≥1). An OR <1 indicates the aid is less effective for people with a history of MHCs than those without, and an OR >1 indicates the aid is more effective.

[3] Bayes factor for the two-way interaction between use of the aid of interest and history of MHCs (0 vs. ≥1), based on expected effect sizes of OR = 0.67 (aid is less effective for users with a history of MHCs) and OR = 1.5 (aid is more effective for users with a history of MHCs). BFs ≥3 can be interpreted as evidence for the alternative hypothesis (i.e., effectiveness of the aid differs according to the user's history of MHCs), BFs ≤1/3 can be interpreted as evidence for the null hypothesis (i.e., effectiveness of the aid does not differ by the user's history of MHCs s), and BFs between 1/3 and 3 suggest that the data are insensitive to distinguish the alternative hypothesis from the null.

Corresponding data with history of MHCs coded as 0, 1, or ≥2 MHCs are provided in **S4 Table**.

successfully than those who did not (OR = 0.41, 95%CI 0.18–0.94). Use of other aids was not significantly associated with odds of quit success, after adjustment.

Tests of interactions showed no statistically significant moderating effect of history of mental health conditions on the effectiveness of any cessation aid (**Table 3**). Bayes factors indicated the data were largely insensitive to distinguish between evidence of moderation and no evidence of moderation, meaning we were unable to rule out potential differences in effectiveness by history of mental health conditions. The only exception was for vaping products, where the data favoured the null (Bayes factor = 0.20), indicating that the effectiveness of vaping products for smoking cessation did not differ significantly between those with and without a history of mental health conditions.

There was no notable difference in the pattern of results when history of mental health conditions was analysed as a 3-level variable (distinguishing between those with none, a single mental health condition, and multiple conditions; **S4 Table**).

## Discussion

Nine in every 20 people who attempted to quit smoking had a history of one or more mental health conditions. Those with a history of mental health conditions were more likely to support their quit attempt with the use of cessation aids. Specifically, they were more likely than

those without a history of mental health conditions to report using vaping products, prescription NRT, or websites, with no significant differences between groups in the use of other aids. After we adjusted for covariates and participants' use of other cessation aids, those who used vaping products, varenicline, or heated tobacco products had significantly higher odds of quitting successfully than those who did not report using these aids. There was little evidence of benefits of using other cessation aids, or that the user's history of mental health conditions moderated the effectiveness of any aid.

Our results echo the findings of the EAGLES trial [38], which found varenicline, bupropion, and nicotine patch were similarly effective for smokers with and without psychiatric disorders, under trial conditions. They also extend existing evidence by covering the whole range of cessation aids used by people who smoke in England and using observational data from people using these aids in the real world, where people do not necessarily receive continued monitoring and support from healthcare professionals.

The real-world effectiveness of varenicline and vaping products are established, having been reported in a number of previous studies [15–24]. In 2019, we used data from the Smoking Toolkit Study to examine the effectiveness of different cessation aids when used in real-world settings [23]. Our results suggested using varenicline or vaping products was associated with the highest odds of success in a quit attempt. When we controlled for use of other aids and other potential confounding variables (e.g., level of dependence), we found that people who used varenicline or vaping products in a quit attempt had 1.82- and 1.95-times higher odds, respectively, of remaining abstinent than those who did not. Our present results, which use data from the same survey but over a different time frame and with additional adjustment for history of mental health conditions, are consistent with these estimates.

However, to our knowledge, this is the first observational study to examine the effectiveness of heated tobacco products, because use of these products until recently has been rare in England [47]. We documented an association between use of heated tobacco products and increased odds of quit success [30]. Our data indicate use of these products in quit attempts remains relatively rare (consistent with low overall prevalence of use among adults in England [47]). As a result, the 95% CI around the estimate was wide and included the possibility of no meaningful difference (lower CI = 1.01), so conclusions may change with more data. It will be important to continue to monitor the effectiveness of heated tobacco products as the number of people using them to quit smoking grows.

The lack of evidence for differential effectiveness of any cessation aid by the user's history of mental health conditions should provide reassurance to people with mental health conditions who want to stop smoking that their condition need not affect their choice of cessation aid. Of note, vaping products were both the most popular aid used by people with and without a history of mental health conditions and one of the most effective. Vaping products were also the only aid for which the data provided clear evidence that effectiveness was not lower for users with a history of mental health conditions (data for the other aids were insensitive).

Our results also suggest that healthcare professionals can base their recommendations for, and prescription of, smoking cessation treatments to people with mental health conditions on evidence of their effectiveness in the general population. Previous research has shown that people with mental health conditions have lower odds of being prescribed varenicline than NRT, despite having greater odds of quitting successfully with varenicline than NRT [54]. Consistent with this, our data show higher prevalence of use of prescription NRT among people with a history of mental health conditions than those without, and significantly higher odds of quit success among users of varenicline, but not NRT. Healthcare professionals may opt to prescribe NRT over varenicline for patients with mental health conditions due to concerns that varenicline may increase the risk of neuropsychiatric adverse events [55]. However, much

evidence of this risk is based on information contained in case reports [55], and a large RCT of the relative neuropsychiatry safety of varenicline compared with nicotine patch and placebo among people with and without psychiatric disorders observed no significant increase in adverse events among those randomised to use varenicline [38]. If the risk of adverse events are similar, offering the more effective treatment (varenicline, once available again) is likely to be the better option.

This study had several limitations. First, questions on mental health conditions relied on self-reports, which may be less accurate than if linked health record data were used. Second, the items assessed ever (as opposed to current) diagnoses. As such, the results cannot tell us whether treatment effectiveness differs according to the user's current mental health status. In addition, those with a history of multiple mental health conditions may also have been diagnosed with these conditions at different points in time, so a history of multiple mental health conditions may not reflect current comorbidity. Third, to boost statistical power for analyses, we grouped together participants reporting any of the mental health conditions we assessed. As the list of conditions was heterogeneous, covering a broad range of conditions from common mental health disorders (e.g., anxiety and depression) to severe mental illness (psychosis), we cannot rule out the possibility that certain conditions may moderate the effectiveness of cessation aids while others do not. Given the low prevalence of most of these conditions within the general population, much larger samples would be required to explore this. Fourth, despite combining all mental health conditions, Bayes factors indicated our data were insensitive to distinguish between evidence of absence of an interaction between aid use and mental health conditions (i.e., mental health conditions moderate treatment effectiveness) and absence of evidence for the majority of aids (all except vaping products). This means we are unable to conclusively rule out there being small to moderate differences in effectiveness by people's history of mental health conditions. Fifth, although we adjusted for a range of potential confounders, there may be residual confounding by other variables not included in our models. Finally, it is possible that smoking cessation treatments offered to people with mental health conditions may differ from the general population, introducing potential selection bias from the treatment provider in addition to the user. Although clinical trials suggest varenicline and bupropion are effective for people with severe mental illness [38, 56], in the real world, people with these conditions are rarely offered these medications by clinicians [54]. Interactions with other medications is an important consideration in deciding on a treatment approach: bupropion is known to interact with other medications (including those indicated for treatment of mental health conditions), which may reduce the effectiveness of one or both treatments [57]. Nonetheless, our data provide useful insights into potential differences in the effectiveness of popular smoking cessation treatments in a real-world setting.

In conclusion, use of vaping products, varenicline, or heated tobacco products in a quit attempt was associated with significantly greater odds of successful cessation, after adjustment for use of other cessation aids and potential confounders. There was no evidence to suggest the effectiveness of any popular cessation aid differed according to the user's history of mental health conditions.

## Supporting information

**S1 Table. Weighted sample characteristics– 3-level mental health variable.**
(PDF)

**S2 Table. Use of cessation aids in the most recent quit attempt by history of mental health conditions– 3-level mental health variable.**
(PDF)

**S3 Table. Use of cessation aids in the most recent quit attempt by history of mental health conditions–by individual mental health condition.**
(PDF)

**S4 Table. Real-world effectiveness of cessation aids for success in stopping smoking and interactions with the user's history of mental health conditions– 3-level mental health variable.**
(PDF)

**S1 Questionnaire. Smoking Toolkit Study questionnaire.**
(DOCX)

## Author Contributions

**Conceptualization:** Sarah E. Jackson, Leonie Brose, Vera Buss, Lion Shahab, Deborah Robson, Jamie Brown.

**Data curation:** Vera Buss, Jamie Brown.

**Formal analysis:** Sarah E. Jackson.

**Funding acquisition:** Leonie Brose, Lion Shahab, Jamie Brown.

**Investigation:** Sarah E. Jackson, Leonie Brose, Vera Buss, Lion Shahab, Deborah Robson, Jamie Brown.

**Methodology:** Sarah E. Jackson, Leonie Brose, Vera Buss, Lion Shahab, Deborah Robson, Jamie Brown.

**Supervision:** Jamie Brown.

**Writing – original draft:** Sarah E. Jackson.

**Writing – review & editing:** Leonie Brose, Vera Buss, Lion Shahab, Deborah Robson, Jamie Brown.

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
