## [Decision Letter · Decision Letter 0]

29 Jan 2024

PMEN-D-23-00045

Moderation of the real-world effectiveness of smoking cessation aids by mental health conditions: a population study

PLOS Mental Health

Dear Dr. Jackson,

Thank you for submitting your manuscript to PLOS Mental Health. After careful consideration, we feel that it has merit but does not fully meet PLOS Mental Health’s publication criteria as it currently stands. Therefore, we invite you to submit a revised version of the manuscript that addresses the points raised during the review process.

Please resubmit your manuscript after the implementing suggested changes below:

We look forward to receiving your revised manuscript.

Kind regards,

Sasidhar Gunturu, MD

Academic Editor

PLOS Mental Health

Journal Requirements:

1. Please amend your detailed online Financial Disclosure statement. This is published with the article. It must therefore be completed in full sentences and contain the exact wording you wish to be published.

a) State the initials, alongside each funding source, of each author to receive each grant. For example: "This work was supported by the National Institutes of Health (####### to AM; ###### to CJ) and the National Science Foundation (###### to AM)."

2. Please declare all competing interests beginning with the statement "I have read the journal's policy and the authors of this manuscript have the following competing interests:"

4. We do not publish any copyright or trademark symbols that usually accompany proprietary names, eg (R), (C), or TM  (e.g. next to drug or reagent names). Please remove all instances of trademark/copyright symbols throughout the text, including ® (Champix®) on page 20.

5. In the online submission form, you indicated that "Data are available from the corresponding author on reasonable request". All PLOS journals now require all data underlying the findings described in their manuscript to be freely available to other researchers, either 1. In a public repository, 2. Within the manuscript itself, or 3. Uploaded as supplementary information.

Additional Editor Comments (if provided):

Reviewers' comments:

Reviewer's Responses to Questions

**Comments to the Author**

1. Does this manuscript meet PLOS Mental Health’s publication criteria? Is the manuscript technically sound, and do the data support the conclusions? The manuscript must describe methodologically and ethically rigorous research with conclusions that are appropriately drawn based on the data presented.

Reviewer #1: Yes

Reviewer #2: Yes

Reviewer #3: Partly

2. Has the statistical analysis been performed appropriately and rigorously?

Reviewer #1: Yes

Reviewer #2: Yes

Reviewer #3: Yes

3. Have the authors made all data underlying the findings in their manuscript fully available (please refer to the Data Availability Statement at the start of the manuscript PDF file)?

Reviewer #1: Yes

Reviewer #2: Yes

Reviewer #3: No

4. Is the manuscript presented in an intelligible fashion and written in standard English?

Reviewer #1: Yes

Reviewer #2: Yes

Reviewer #3: Yes

5. Review Comments to the Author

Reviewer #1: Dear Jackson et al,

It has been interesting to read the paper. I know this is well written and has important implication for addiction psychiatry and where patients with mental health disorders have comorbid Tobacco Use Disorder. However, I know we can some confounding factors like race, place of residence, migration status and educational level. Can you explain if you considered those potential confounding factors and how you addressed this issue.

Otherwise, I would accept this paper to be published.

Reviewer #2: I think this is a great study, thank you for letting me review it. Just a few comments:

1. Consider replacing "stopping smoking" to "smoking cessation"

2. Allen Carr's Easyway method is not familiar to me - consider giving an explanation for non-UK readers about what this is

3. Page 5 lines 89-90: However, a recent secondary analysis of another RCT suggested bupropion was less

effective for smokers with depressive symptoms than those without." This is a little bit confusing. I think you should flesh out an explanation about it more

4. There is some confusion about whether you are referring to sex or gender in your analysis. You say "women and non-binary" which would be gender but in the table it says "sex"

5. I think it would be awesome to see the p-values for table 2 as well as including the numbers for one type of therapy only compared to multiple, as it is unclear how many people had multiple versus just one

6.

Reviewer #3: Summary

The authors present a population-based investigation in England that compares the real-world effectiveness of smoking cessation aids in patients with and without co-occurring mental health conditions.

Introduction

The Introduction draws no clear conclusion as to whether previous investigators have addressed the same question, namely “to examine whether real-world effectiveness of popular smoking cessation aids differ between users with and without a history of mental health conditions.”

• Lines 87-90 cite reference 37, concluding that a large RCT showed similar efficacy for smokers with and without psychiatric disorders.

• The Discussion also cites the EAGLES trial, with similar findings.

Are their other published studies supporting or refuting same? Since the association with psychiatric conditions constitutes the Objective of the investigation, further discussion would prove useful. Establishing whether or not the findings are a) unique and/or b) groundbreaking would significantly strengthen the presentation.

Although the manuscript provides a statistical analysis of smoking cessation aids, the authors should consider also strengthening the investigation by characterizing the underlying epidemiologic data in England pertaining to smoking and mental conditions, ie, the percentage(s) of patients with underlying mental conditions who smoke vs those without underlying mental conditions who smoke.

Lines 70-90: These two paragraphs (excepting Lines 90-91) intermix two related issues that merit discussion separately.

• The first sentence of the first paragraph (Lines 70-71) references the authors’ prior work while the remainder of the paragraph transitions to findings—which are more related to cessation aids than to mental health conditions. Moving the cessation aid findings to the Discussion would strengthen the Introduction, which introduces the rationale for the investigation of mental health issues.

• Likewise, although the second paragraph (Lines 80-90) addresses mental health conditions, the content appears to blend discussion of the rationale for the investigation (appropriate to the Introduction) with additional material (more appropriate for the Discussion).

Lines 90-91 assert that “observational data are required to shed light on any differences in treatment effectiveness in real-world settings”. Although commonly articulated, such a perspective is not necessarily supported by the medical literature examining the topic in more detail:

A comparison of observational studies and randomized, controlled trials. New Engl J Med 2000;342:1878-86

Acknowledgment and consideration would serve to strengthen the Conclusions.

Methods and Materials

Although the manuscript describes several individual aspects of the Smoking Toolkit study in multiple places, readers not familiar with the Toolkit (particularly those residing outside England) would benefit from a more detailed. Specifically:

• What is the overlap, if any, between the Toolkit and the reported trial?

• Lines 115-117 refer to face-to-face vs telephone interviews. What was the relationship between the national database and the interviews? Were they actually Toolkit interviews performed as part of the ongoing Toolkit survey: Or were they in addition to the Toolkit data acquisition (ie, layered on top).

• It appears that one or more of the authors may have relationships with the Toolkit effort. If so, could you please elaborate on the overlap?

I am not concerned about the propriety of such a connection; just suggesting that the reader needs a clearer picture of the interrelationships, if any.

The methodology needs clarification of the non-contiguous time frames for the data collected.

• If the data were collected monthly between 2016-17 and 2020-23, can the authors please explain the reason for excluding the data for 2018 and 2019?

• Are the data collected monthly on an ongoing basis regardless of the time frames stated in the manuscript?

• Why did the authors choose non-contiguous time frames?

Some of these concerns may be answered by addressing the questions raised in the previous paragraph (ie, Toolkit).

Line 175: For those who may not be familiar with ‘Stoptober’; a short description and/or citation would prove helpful.

Discussion

The latter portion of the manuscript, starting with the Discussion, lacks line numbers.

The moderator of “diagnosed mental health conditions” contains a broad spectrum of conditions ranging from neuroses (mild) to psychoses (severe). While the authors conclude that the effectiveness of three cessation aids did not vary by presence or absence of the listed mental health conditions, they do not address whether additional findings might be unmasked by more detailed analysis of the nature (and implied severity) of individual underlying conditions, ie, anxiety vs psychosis. Such consideration would not only strengthen their “real world” conclusions—it might also lead to important new, yet otherwise unsuspected, findings that serve to further knowledge with regard to the study objective. Specifically, although “mental health conditions” as a group do not appear to affect the overall outcome, do any of the listed individual “mental health conditions” achieve statistical significance separately?

6. PLOS authors have the option to publish the peer review history of their article (what does this mean?). If published, this will include your full peer review and any attached files.

**Do you want your identity to be public for this peer review?** For information about this choice, including consent withdrawal, please see our Privacy Policy.

Reviewer #1: No

Reviewer #2: No

Reviewer #3: No

[NOTE: If reviewer comments were submitted as an attachment file, they will be attached to this email and accessible via the submission site. Please log into your account, locate the manuscript record, and check for the action link "View Attachments". If this link does not appear, there are no attachment file

---

## [Editor Report · Decision Letter 1]

8 Mar 2024

Moderation of the real-world effectiveness of smoking cessation aids by mental health conditions: a population study

PMEN-D-23-00045R1

Dear Dr. Jackson,

We are pleased to inform you that your manuscript 'Moderation of the real-world effectiveness of smoking cessation aids by mental health conditions: a population study' has been provisionally accepted for publication in PLOS Mental Health.

Best regards,

Sasidhar Gunturu, MD

Academic Editor

PLOS Mental Health